# Jasmone Is a Ligand-Selective Allosteric Antagonist of Aryl Hydrocarbon Receptor (AhR)

**DOI:** 10.3390/ijms242115655

**Published:** 2023-10-27

**Authors:** Radim Vrzal, Adéla Marcalíková, Kristýna Krasulová, Lenka Zemánková, Zdeněk Dvořák

**Affiliations:** Department of Cell Biology and Genetics, Faculty of Science, Palacky University, Slechtitelu 27, 783 71 Olomouc, Czech Republic

**Keywords:** jasmone, AhR, CYP1A1, LS180, HepG2, ChIP

## Abstract

Herbal extracts represent a wide spectrum of biologically active ingredients with potential medical applications. By screening minor constituents of jasmine essential oil towards aryl hydrocarbon receptor (AhR) activity using a gene reporter assay (GRA), we found the antagonist effects of jasmone (3-methyl-2-[(2Z)-pent-2-en-1-yl]cyclopent-2-en-1-one). It inhibited 2,3,7,8-tetrachlorodibenzo-p-dioxin (TCDD)-, benzo[a]pyrene (BaP)-, and 6-formylindolo[3,2-b]carbazole (FICZ)-triggered AhR-dependent luciferase activity in a concentration-dependent manner. However, the inhibition differed markedly between TCDD, BaP, and FICZ, with the latter being significantly less inhibited. The dose-response analysis confirmed an allosteric type of AhR antagonism. Furthermore, jasmone efficiently inhibited AhR activation by AhR agonists and microbial catabolites of tryptophan (MICTs). TCDD- and FICZ-inducible CYP1A1 expression in primary human hepatocytes was inhibited by jasmone, whereas in the human HepG2 and LS180 cells, jasmone antagonized only TCDD-activated AhR. Jasmone only partially displaced radiolabeled TCDD from its binding to mouse Ahr, suggesting it is not a typical orthosteric ligand of AhR. TCDD-elicited AhR nuclear translocation was not affected by jasmone, whereas downstream signaling events, including the formation of the AhR:ARNT complex and enrichment of the CYP1A1 promoter, were inhibited by jasmone. In conclusion, we show that jasmone is a potent allosteric antagonist of AhR. Such discovery may help to find and/or clarify the use of jasmone in pharmaco- and phytotherapy for conditions where AhR plays a key role.

## 1. Introduction

The aryl hydrocarbon receptor (AhR) is a ligand-activated transcription factor that belongs to the Per-ARNT-Sim (PAS) family of proteins, which integrates stimuli from the environment [1]. Initially, it was identified as a mediator of the toxic action of 2,3,7,8-tetrachlorodibenzo-*p*-dioxin (TCDD), a by-product of herbicide 2,4,5-trichlorofenoxyacetic acid synthesis [2]. The receptor is expressed in many tissues, with the highest amount detected in the lungs, placenta, and spleen [3]. This suggests its chemoprotective role in tissues that interact with environmental stimuli.

In the absence of a ligand, AhR resides in the cytosol of the cells in a complex with two HSP90s and AIP and p23 proteins [4,5]. After ligand binding, AhR translocates into the nucleus to heterodimerize with the AhR nuclear translocator (ARNT), and they bind to the consensus sequences known as dioxin/xenobiotic response elements (DREs/XREs). This can be found in the promoters of many genes whose products participate in biotransformation (e.g., CYP1A1/1A2/1B1, UGT1A6, GSTA1), cell tight junction coherence, apoptosis, cell cycle, immune regulation, etc. [6,7,8]. The signaling is terminated by the proteasomal degradation of AhR [9].

Among AhR’s ligands are hydrophobic low-molecular-weight compounds of exogenous origin, from planar hydrocarbons, considered originally “classic ligands”, up to natural compounds like polyphenols or metabolites of arachidonic acid [10]. All these compounds can act as agonists and antagonists of AhR [11]. Among the endogenous ligands and activators of AhR, different derivatives of a compound named indole, like 6-formylindolo[3,2-b]carbazol (FICZ), 2-(1H-Indol-3-ylcarbonyl)-4-thiazolecarboxylic acid methyl ester (ITE), tryptamine, indirubin, or indigo, have been identified [12,13,14]. Unlike “classic ligands” that are usually persistent, the latter mentioned are usually rapidly metabolized and are suggested and considered for AhR-mediated pharmaco-/chemotherapy [15]. Therefore, the search for “non-classical”, non-persistent ligands of AhR among naturally occurring compounds is of interest.

A specific topic regarding AhR ligands is represented by those that can be classified as antagonists. These compounds possess the ability to inhibit agonist-induced AhR activation. Recently, their popularity increased upon finding that AhR activation can be induced by infections with coronaviruses, like HCoV-229E (common cold) and SARS-CoV-2 (COVID-19), and that the use of AhR antagonists can boost antiviral immunity [16]. Thus, both types of ligands, agonists and antagonists, may find appropriate use in human therapy for different pathological states.

An enormous source of compounds with either agonist or antagonist activity comes from nature. In folk medicine throughout history, essential oils (EOs) have been used. These concentrated hydrophobic liquids containing volatile aroma compounds from plants have found new popularity in aromatherapy, cosmetics, the food industry, and gastronomy. They are obtained mainly by distillation, steam distillation, cold pressing, or solvent extraction from the plants [17]. Due to their increased use in gastronomy, the dietary intake of EOs and their constituents is significant and deserves attention in terms of pharmacology and toxicology. This was the goal in a recent publication, in which the interaction of selected EOs with AhR signaling was investigated [18]. By employing a reporter gene assay in stably transfected HepG2 cells (AZ-AhR cells), it was demonstrated that 4 EOs displayed properties of full agonists, 5 EOs of partial agonists, 8 EOs of antagonists, and 14 EOs did not display any AhR activity. The induction of the most sensitive AhR target gene, CYP1A1, was weak and reached less than 10% of induction by TCDD. To uncover the AhR-active constituents of EOs, the effects of major ingredients (>10%) were tested. In the group of EOs acting like full agonists (cumin, jasmine, vanilla, bay leaf), none of the tested components activated AhR (measured as luciferase activity) comparably either with EOs alone or a mix composed of these major (>10%) constituents [18].

Therefore, in the current paper, we decided to subject the minor constituents of jasmine essential oil (1–10%) to screening to discover naturally based, chemically defined compounds that could be further tested in a relevant pathological context in which AhR plays a key role. Among these, we discovered the strong antagonistic activity of jasmone.

## 2. Results

### 2.1. Jasmone Is an AhR Antagonist

Minor jasmine oil constituents (between 1 and 10%) were screened for AhR activity in the reporter hepatoma cell line AZ-AhR for 4 h (Figure 1A). Among the 14 tested compounds, only the microbial catabolite of tryptophan (MICT), indole, induced AhR-dependent luciferase activity significantly and robustly. Four compounds (eugenol, farnesene, jasmone, and linalool) decreased AhR-dependent luciferase activity in a concentration-dependent manner, and other compounds (squalene, epoxysqualene, benzyl alcohol, geranyl linalool, isophytol, methyl linoleate, α-linolenic acid, palmitic acid, and phytyl acetate) had no impact on luciferase activity. Among the four compounds displaying antagonist-like behaviors, the lowest inhibitory IC50 value together with maximal luciferase activity suppression was observed for jasmone. For that reason, only jasmone was further subjected to a detailed molecular analysis of AhR activity.

One of the reasons for the jasmone-triggered decline in basal AhR-mediated luciferase activity could be either luciferase activity inhibition or cytotoxicity. The first one was ruled out by measuring the luciferase activity of TCDD-treated AZ-AhR cellular lysate with an increasing concentration of jasmone (Appendix A). To address the second question, we used two cytotoxicity assays, MTT and the crystal violet assay (CVA). A significant impact was observed on mitochondrial metabolic activity in HepG2 but not in LS180 cells (Appendix A). This decline in the MTT assay is consistent with a previous report about the impact of jasmonates on mitochondrial respiration [19]. On the other hand, we found no impact on proliferation (reflected by CVA) at 4 h for both cell lines (HepG2, LS180) and only a mild impact at 24 h in HepG2 (from which the AZ-AhR cell line is derived) but not in LS180 cells (Appendix A). In conclusion, the jasmone-triggered decline in AhR-dependent luciferase activity is not due to the cytotoxicity of jasmone.

Given that a fresh cultivation medium is capable of AhR activation that ceases with longer incubation, it raised the question of whether rapidly metabolized AhR activators, such as MICTs, could be antagonized by jasmone. The ability of MICTs to activate AhR was deeply investigated by our group recently [20,21]. By using a single concentration of each MICT (10 μM), we found that jasmone inhibited AhR-dependent luciferase activity in a concentration-dependent manner, with the lowest IC50 values found for indole-3-acetic acid (IAA), indole-3-lactic acid (ILA), tryptamine (TRY), indole-3-propionic acid (IPA), indole-3-aldehyde (IA), and indole-3-ethanol (IET) (Figure 1B). Thus, despite the limited number of serum components used in a single concentration without reflection of the mixture effect, it is likely that the jasmone action displayed at the beginning of the screening (Figure 1A) in fresh medium might have at least partially contributed to the presence of microbial catabolites of tryptophan (MICTs), which are present in serum [22,23].

As a consecutive step in our effort to characterize jasmone’s antagonistic activity, we tested jasmone against high-affinity, full AhR agonists, specifically TCDD, BaP, and FICZ. These compounds were applied at fixed concentrations corresponding to their EC80s, which were determined using a reporter gene assay in AZ-AhR cells for 4 and 24 h of incubation [24]. While jasmone antagonized both TCDD and BaP concentration-dependently with very similar IC50s (9–11 µg/mL) (Figure 1C), FICZ-induced luciferase activity was affected less intensively after 4 h compared to TCDD/BaP (IC50 = 33 ± 7 µg/mL), and it almost disappeared after 24 h (IC50 = 102 ± 13 µg/mL). Since both ligands, TCDD and BaP, interact with the same amino acid residues in the ligand-binding pocket (LBD) of AhR [25] and the induced luciferase activity was inhibited similarly, we further excluded BaP from our investigations.

Our last step using the reporter gene assay was to clarify the nature of the antagonistic behavior displayed by jasmone. We measured dose-response curves for TCDD and FICZ in combination with increasing jasmone concentrations in depleted (from 2-day-old cultures) medium to eliminate the activity of the serum components that activate AhR, e.g., MICTs. This analysis resulted in an increasing EC50 for TCDD but not for FICZ (Figure 1D) and a gradual decrease in maximal response (E_MAX_) for both compounds. This suggested an allosteric type of antagonism for jasmone.

### 2.2. Jasmone Suppresses Ligand-Triggered AhR-Regulated Genes

After intensive investigation of jasmone’s antagonistic activity by means of reporter gene assays, we examined AhR target gene expression in HepG2 and LS180 cells. The latter cell line was selected to demonstrate the impact of jasmone on AhR signaling in cells of intestinal origin. We monitored the expression of the most sensitive indicator of AhR activation, CYP1A1. By using TCDD (13.5 nM) and FICZ (22.6 µM), we observed fold inductions of roughly 320 and 415, respectively, after 24 h in HepG2 cells (Figure 2A). Jasmone alone only insignificantly increased CYP1A1 mRNA in HepG2 cells, and the co-incubation of the AhR ligands with jasmone resulted in a concentration-dependent decrease in CYP1A1 mRNA for TCDD but not FICZ. A similar profile was further observed at the protein and catalytic activity levels (Figure 2B,C). In LS180 cells, the profile was a little bit different since TCDD-inducible CYP1A1 mRNA was effectively inhibited at 4 h but not at 24 h of incubation (Figure 2D). No effect of jasmone on FICZ-inducible CYP1A1 mRNA was observed at any time tested. Similarly, TCDD- but not FICZ-induced CYP1A1 protein level was suppressed by jasmone at 4 and 24 h in the intestinal LS180 cell line (Figure 2E). Hence, to a certain extent, jasmone exhibited similar behavior to that observed in HepG2 cells. In addition, by employing primary cultures of human hepatocytes (PCHHs), we found that jasmone inhibited TCDD- and FICZ-induced CYP1A1 mRNA (Figure 2F). Moreover, other measured AhR-responsive genes, like CYP1B1, AhRR, and CYP1A2, were mostly inhibited with a similar profile to that of CYP1A1 (Appendix A).

### 2.3. Jasmone Affects the Process of Dimerization in Ligand-Triggered AhR Signaling

In the following experiments, we sought to determine where the interference of jasmone with TCDD-/FICZ-triggered AhR signaling occurs. First, we employed a ligand-binding assay to demonstrate if jasmone acts as an AhR ligand by competing with [^3^H]-TCDD-specific binding to mouse AhR (mAhr). This choice was based on the fact that human AhR is quite labile and degrades quickly [26]. While the positive control (FICZ) decreased the radioactivity of [^3^H]-TCDD, the unspecific glucocorticoid dexamethasone had no effect (Figure 3A). Jasmone (100 μg/mL~609 μM) slightly decreased the radioactivity of [^3^H]-TCDD on average by 40%. This concentration, which did not reach IC50, was 10 times higher than the IC50 for jasmone antagonism in the cell-based assay when co-incubated with TCDD in AZ-AhR cells (Figure 1C). This lack of ability to displace [^3^H]-TCDD suggested either the impact of jasmone on mAhR conformation or species differences between mouse and human AhRs.

It is known that mAhR and hAhR have the same amino acid (AA) residues in the appropriate positions of LBD. However, certain AA residues in the surrounding regions differ, which may contribute to species-specific antagonism, as described previously [27,28]. To ensure that the results of the ligand-binding assay performed with mouse AhR did not compromise the results of the effect of jasmone on human AhR activity in the cell-based assay, we generated a stably transfected mouse hepatoma cell line (Hepa1c1c7 cells) responsive to AhR and named it Aherepa. A closer characterization of this cell line revealed picomolar EC50s for TCDD and FICZ at 4 h of treatment but picomolar and nanomolar EC50s for TCDD and FICZ, respectively, after 24 h (Appendix A). Thus, the murine hepatoma cell line Hepa1c1c7 acts similarly to human HepG2, i.e., FICZ potency is diminished with time by approximately three orders of magnitude. However, jasmone was able to effectively inhibit TCDD (using EC80 for 4 h~300 pM) but less effectively inhibit FICZ (using EC80 for 4 h~80 pM), with IC50s of approximately 40 ± 8 µg/mL and more than 100 µg/mL, respectively (Appendix A). The above results confirmed that the impact of jasmone is not species-specific, and it does antagonize human as well as murine AhRs despite the weak impact on the displacement of radiolabeled TCDD in the ligand-binding assay.

Secondly, after ligand binding, the next step in AhR signaling is the translocation to the nucleus. By employing immunofluorescence, we found that jasmone did not prevent AhR nuclear translocation in the presence of TCDD or FICZ in LS180 cells after 90 min of incubation, as the number of AhR-positive nuclei remained equal (Figure 3B and Appendix A).

Thirdly, the heterodimerization of AHR with ARNT, which proceeds before DNA binding and the start of transcription, was evaluated. By using the co-immunoprecipitation assay, we found that jasmone disrupted TCDD- but not FICZ-induced AhR:ARNT complex formation in LS180 cells after 90 min of incubation (Figure 3C). The unequal AhR loading seen in WCE is likely due to rapid ligand-elicit AhR protein degradation in the intestinal LS180 cells (Appendix A). Interestingly, TCDD- but not FICZ-induced AhR degradation was abolished by jasmone.

Fourthly, and consistent with the observed abolished heterocomplex formation for TCDD but not for FICZ, disrupted binding to the CYP1A1 promoter both in HepG2 and LS180 cells was observed for the combination of jasmone and TCDD but not jasmone and FICZ (Figure 3D).

## 3. Discussion

The aryl hydrocarbon receptor (AhR) was long considered a xenosensor that helped to eliminate undesirable toxicological burdens. However, in the last two decades, the view has changed due to the observation that usually natural compounds transiently activating AhR do not represent any sign of toxicity observed by dioxin-like compounds and that this activation is physiologically important and can bring benefits for health and proper development (reviewed in [15]). Moreover, a search for a particular compound that acts as an antagonist was recently demonstrated since it may help to boost antiviral immunity as AhR is present in immune cells as well. Thus, the identification of compounds that deactivate AhR may bring new advances in pharmacology as well as help to better understand the protective effects of some dietary habits. However, deactivation must be considered in the context of already present ligand(s), which activate AhR.

At the beginning, we found that four compounds identified in jasmine essential oil (namely, eugenol, farnesene, jasmone, and linalool) displayed antagonist-like behavior in a gene reporter assay monitoring AhR activity, with jasmone displaying the most intensive effect.

Among those four candidates, eugenol’s activity was a little bit surprising since, together with isoeugenol, it was demonstrated to induce rapid translocation of AhR, increased expression of CYP1A1, and the inhibition of human HaCaT keratinocyte proliferation [29]. This would classify eugenol as an agonist. However, since the longer incubation time (24 h) of AZ-AhR cells with eugenol resulted in the induction of AhR-mediated luciferase activity (personal observation), it is clear that eugenol fulfills the definition of a partial agonist. This finding may contribute to the explanation of why clove essential oil, which contains eugenol as a volatile component, is useful for the treatment of infections, lesions, and inflammatory conditions in the skin.

Jasmone is an organic compound that is produced by some plants via the decarboxylation of jasmonic acid. It is appreciated as a fragrance and as a plant-derived signal molecule controlling pollination. As a component of perfumes and essential oils, it comes into contact with humans. Thus, it can affect certain signaling pathways in the skin and the gastrointestinal and pulmonary tracts. Surprisingly, there are not many data on the biological effects of jasmone in humans or human “in vitro” cellular models.

While some studies described the impact on cancer cells, like non-small-cell lung cancer lines [30], breast cancer cell lines [31], or hormone-independent prostate cancer cells [32], in terms of decreased proliferation and apoptosis induction, no study investigated if and how jasmone affects ligand-activated transcription factors, particularly in hepatic or intestinal cell lines. Nevertheless, we are consistent in terms of decreased mitochondrial respiration (Appendix A) and proliferation (Appendix A) for hepatoma HepG2 cells despite the lower concentration of jasmone used (100 μg/mL, approx. 609 μM) in contrast to the above-mentioned studies (1–3 mM jasmone). Moreover, the type of tissue is important, as no significant change was observed for LS180 cells (Appendix A).

All these data emphasize the importance of place, dose, and time. Certainly, there are differences between the intestinal and hepatic cell lines, as jasmone was more effective in HepG2 than LS180 cells in terms of TCDD-induced CYP1A1 mRNA (compare Figure 2A vs. Figure 2D) and, particularly, of CYP1A1 protein and catalytic activity when the intensive impact of jasmone was observed in lower concentrations (1–10 μg/mL~6–61 μM) (Figure 2B,C). In addition, the effect of jasmone is persuasively observed in short incubation times (in LS180 cells only), which may suggest the metabolic conversion of jasmone in LS180 cells. On the other hand, the most metabolically competent model used, primary cultures of human hepatocytes, had probably minimal impact on jasmone metabolism since it antagonized both TCDD and FICZ, as was observed for AhR target genes, CYP1A1/1A2/1B1. The possible reason for jasmone-antagonized FICZ-induced CYPs in PCHH but not HepG2 or LS180 may be the high constitutive expression of CYP1A2, which contributes to FICZ metabolism. In cancerous cells, CYP1A2 and CYP1A1 are usually absent, particularly CYP1A1, which is highly inducible and inhibited by nanomolar concentrations of FICZ [33]. Thus, a high concentration of FICZ may inhibit newly formed CYP1A1 from the very beginning, and this may contribute to the absence of jasmone’s inhibiting effects in HepG2 as well as LS180 cells when combined with FICZ. All these aspects should be considered if jasmone is to be considered for any kind of therapeutic application.

Since we demonstrated that jasmone antagonized ligand-triggered AhR activity, it may be of particular interest if its application might help to boost antiviral immunity, e.g., after exposure to viruses like HCoV-229E (common cold) or SARS-CoV-2 (COVID-19). This might be a challenge since proper dose, place of application (mouth or nasal spray), and frequency could demonstrate the usefulness of our discovery under in vivo conditions.

Jasmone can be found in fragrances and essential oils. The latter, particularly, is the source where we initially started the screening “https://pranarom.fr/search?q=jasmine&type=product (accessed on 17 September 2023)”. In this particular flask, it was present in about 4.09% (*v*/*v*) and both forms, i.e., as cis- and trans-jasmone. By using basic mathematical calculations, we determined that there was approximately 39 mg/mL of jasmone in the flask. In one drop, which is approximately 50 μL, there was 1.94 mg of jasmone. This corresponded roughly to a concentration of 236 mM, and even if the drop were diluted 1000 times, it would still be in the concentration range effective for AhR antagonism. This observation could help to shed light on some of the advantages associated with aromatherapy. Nevertheless, many bioactive components in essential oils may either enhance or compromise jasmone’s effect.

Another aspect lies in the application of essential culinary oils, which may contain jasmone as well. Therefore, an impact on the gastrointestinal tract can be expected. As demonstrated, jasmone effectively inhibited AhR activity triggered by microbial catabolites of tryptophan (MICTs) in relatively low concentrations. Consequently, it is reasonable to anticipate a degree of AhR inhibition when dealing with pure essential oils containing jasmone. Nevertheless, this scenario appears less likely when working with culinary essential oils, where the concentration, combined with other constituents and the presence of a meal, is unlikely to pose any harm.

However, its antagonistic action may represent a factor that promotes intestinal inflammation. Similarly, in a recent study, it was demonstrated that herbicide propyzamide acted as an AhR antagonist and, together with TNBS (2,4,6-trinitrobenzene sulfonic acid, a chemical inductor of colitis in mice), boosted intestinal pathologies in zebrafish [34]. Therefore, it is conceivable that individuals with pre-existing intestinal dysbiosis, such as those suffering from inflammatory bowel disease (IBD), might experience some degree of adverse effects.

We have recently published that essential oils from dill, cumin, and blackberry have antagonistic effects on AhR and that carvones, the main constituents of these essential oils, are responsible for AhR antagonism. Thus, a comparison with this recently identified allosteric AhR antagonist is suggested [24]. A deep molecular analysis revealed allosteric binding of carvone to AhR, with tyrosine at position 76 being likely the key amino acid interacting with carvone. Despite different chemical formulas (C_10_H_14_O for carvone vs. C_11_H_16_O for jasmone) and structures (substituted cyclohexanone for carvone vs. substituted cyclopentenone for jasmone), the molecular effects of these two compounds displayed very similar behavior in the tests and concentrations used (carvone inhibited up to 1000 μM; jasmone up to 100 μg/mL~609 μM). Thus, it appears that jasmone is another allosteric antagonist of AhR based on the insufficient replacement of radiolabeled TCDD in ligand-binding assays, the lack of ability to prevent ligand-triggered nuclear translocation of AhR, and disrupted AhR:ARNT heterocomplex formation. Therefore, a similar use of jasmone can be anticipated, as in the case of carvone.

In conclusion, an essential oil constituent, jasmone, is capable of antagonizing “classic” AhR ligands, like dioxins and aromatic hydrocarbons, but antagonizes the endogenous ligand FICZ less effectively. This may be due either to the presence of supraphysiological FICZ concentrations or the specific interaction of jasmone with AhR amino acid residues that are essential for the heterocomplex formation of TCDD but not FICZ. The results presented here may be the starting point for the chemotherapeutic use of jasmone as an AhR antagonist.

## 4. Materials and Methods

### 4.1. Chemicals

6-formylindolo[3,2-b]carbazole (FICZ), dimethylsulfoxide (DMSO), benzo[a]pyrene (BaP), hygromycin B, benzyl alcohol, epoxysqualene, Dulbecco’s modified Eagle’s medium (DMEM), fetal bovine serum (FBS), indole-3-aldehyde (IA), and indole-3-ethanol (IET) were purchased from Merck/Sigma (Prague, Czech Republic). Jasmone, alpha-linoleic acid, eugenol, farnesen, geranyllinalool, indole, isophytol, linalool, methyllinolenate, palmitic acid, phytyl acetate, squalene, and other MICTs were purchased from Santa Cruz Biotechnology (Santa Cruz, CA, USA). 2,3,7,8-tetrachlorodibenzo-*p*-dioxin (TCDD) was purchased from Ultra Scientific (North Kingstown, RI, USA). LightCycler 480 Probes Master and UPL probes were obtained from Roche Diagnostic Corporation (Intes Bohemia, Praha, Czech Republic). Luciferase lysis buffer was purchased from Promega (Madison, CA, USA). DAPI (4′,6-diamino-2-phenylindole) was obtained from Serva (Heidelberg, Germany). [^3^H]-TCDD was purchased from American Radiolabeled Chemicals. Bio-Gel^®^ HTP Hydroxyapatite was obtained from Bio-Rad Laboratories, and 2,3,7,8-tetrachlorodibenzofuran (TCDF) from Ambinter (Orleáns, France). The oligonucleotide primers used in the RT-PCR reactions were purchased from Generi Biotech (Hradec Kralove, Czech Republic), and their sequences and corresponding UPL probe numbers are depicted in Table 1. All other chemicals were of the highest quality commercially available.

### 4.2. Cell Cultures

Human Caucasian hepatocellular carcinoma HepG2 cells (ECACC No. 85011430) and human colon adenocarcinoma LS180 cells (ECACC No. 87021202) were cultured in Dulbecco’s modified Eagle’s medium (DMEM) supplemented with 10% of fetal bovine serum, 4 mM L-glutamine, 1% non-essential amino acids, and 1 mM sodium pyruvate. Cells were maintained at 37 °C and 5% CO_2_ in a humidified incubator.

### 4.3. Human Hepatocytes

The primary cultures of human hepatocytes (PCHHs) used in this study were short-term primary human hepatocytes in monolayer batches Hep2201029 (male, 87 years, unknown ethnicity) and Hep2201032 (male, 43 years, Caucasian), which were purchased from Biopredic International (Rennes, France). PCHHs were maintained in serum-free cultivation medium as described elsewhere [35].

### 4.4. Immunostaining of AhR Translocation

LS180 cells (60,000/well) were plated on 8-well chambered slides (Ibidi GmbH, Planegg, Germany), allowing them to grow overnight. Afterwards, cells were treated for 90 min with 100 µg/mL of jasmone in the presence of 13.5 nM TCDD and 22.6 µM FICZ, along with positive (13.5 nM TCDD, 22.6 µM FICZ) and negative (0.1% *v*/*v* DMSO) controls. Cells on the slides were fixed with 4% formaldehyde in PBS for 30 min, then permeabilized with 0.1% Triton-X-100 in PBS for 5 min, and blocked with 3% BSA in PBS for 45 min. Then, cells were incubated with Alexa Fluor 488-labeled primary antibody against AhR (Santa Cruz Biotechnology, Dallas, TX, USA) diluted 1:500 in 0.5% BSA at 4 °C overnight. The next day, nuclei were stained with DAPI, and cells were enclosed in VectaShield Antifade Mounting Medium (Vector Laboratories Inc., Peterborough, UK). Visualization of AhR translocation into the nucleus was performed using the IX73 fluorescence microscope (Olympus, Tokyo, Japan). Each experiment was performed in duplicate, and two independent measurements were performed for each replicate.

### 4.5. Co-Immunoprecipitation (Co-IP)

Formation of AhR-ARNT heterodimer was studied in cell lysates from intestinal LS180 cells, which were incubated with vehicle (DMSO; 0.1% *v*/*v*; EtOH abs.; 0.1% *v*/*v*) and TCDD (13.5 nM) or FICZ (22.6 µM) in the presence or absence of jasmone (100 µg/mL) for 90 min at 37 °C. Pierce™ Co-Immunoprecipitation Kit (Thermo Fisher Scientific, Waltham, MA, USA) with a covalently coupled AhR antibody (mouse monoclonal, sc-133088, A-3, Santa Cruz Biotechnology, Dallas, TX, USA) was used according to the manufacturer’s instructions. Eluted protein complexes and parental total lysates were resolved on SDS-PAGE gels, followed by Western blot and immunodetection with ARNT1 antibody (mouse monoclonal, sc-17812, G-3, Santa Cruz Biotechnology). Chemiluminescent detection was performed using horseradish peroxidase-conjugated anti-mouse secondary antibody (7076S, Cell Signaling Technology, Danvers, MA, USA) and WesternSure^®^ PREMIUM Chemiluminescent Substrate (LI-COR Biotechnology, Lincoln, NE, USA) in a C-DiGit^®^ Blot Scanner (LI-COR Biotechnology).

### 4.6. SDS-PAGE and Western Blotting

The complete procedure, from protein extract preparation to final protein detection, was described elsewhere [36]. The primary antibodies used in this project were β-actin (8H10D10, mouse monoclonal) from Cell Signaling Technology and AhR (sc-133088, mouse monoclonal) and CYP1A1 (sc-393979) from Santa Cruz Biotechnology.

### 4.7. Reporter Gene Assays (RGAs)

AZ-AHR cells [37] were incubated for 4 h with a vehicle (EtOH; 0.1% *v*/*v*) and increasing concentrations of the tested minor jasmine essential oil constituents, or for 4 and 24 h with a vehicle or jasmone (1–100 g/mL) in the presence of TCDD (20 nM/13.5 nM), FICZ (8 nM/22.6 μM), or BaP (7 μM/15.8 μM). Two concentrations of the appropriate ligand represent the calculated EC80 for 4 or 24 h of incubation with a given ligand alone [24].

After incubation, cells were lysed, and luciferase activity was measured using a Tecan Infinite M200 plate luminometer (Schoeller Instruments, Praha-Kunratice, Czech Republic). Experiments were performed in at least five consecutive cell passages, with treatments performed in quadruplicates (technical replicates). The values of half-maximal inhibitory concentration (IC_50_) and efficiency concentrations (EC_50_, EC_80_) were calculated in GraphPad Prism 6.

### 4.8. Chromatin Immunoprecipitation Assay (ChIP)

HepG2 cells (4 million) and LS180 cells (5 million) were seeded in a 60 mm dish in Dulbecco’s modified Eagle’s medium (D6546; Sigma Aldrich, St. Louis, MI, USA); the following day, they were incubated with negative control (0.1% DMSO/EtOH), TCDD (13.5 nM), and FICZ (22.6 µM) in the absence or presence of jasmone (100 µg/mL) for 90 min at 37 °C. The rest of the procedure is described elsewhere [38].

### 4.9. RNA Isolation, Reverse Transcription, and PCR

Total RNA isolation, reverse transcription, and qPCR were described elsewhere [39]. The levels of all mRNAs were determined using primers and the Universal Probe Library (UPL; Roche Diagnostic Corporation, Prague, Czech Republic) described elsewhere [40]. The measurements were performed in triplicate. Gene expression was normalized to glyceraldehyde-3-phosphate dehydrogenase (GAPDH) as a housekeeping gene. Data were processed by the delta–delta method.

### 4.10. Radioligand Binding Assay

Cytosolic protein from murine hepatoma Hepa1c1c7 cells (2 mg/mL) was incubated for 2 h at room temperature in the presence of 2 nM [^3^H]-TCDD with jasmone (100 µg/mL), FICZ (100 nM; positive control), dexamethasone (100 nM; negative control), or a vehicle (DMSO; 0.1% *v*/*v*; corresponds to specific binding of [^3^H]-TCDD = 100%). The hydroxyapatite-binding protocol and scintillation counting determined ligand binding to the cytosolic proteins. Specific binding of [^3^H]-TCDD was determined as a difference between total and non-specific (TCDF; 200 nM) reactions. Two independent experiments were performed, and the incubation and measurements were carried out in triplicate in each experiment (technical replicates).

### 4.11. P450-Glo CYP1A1 Assay

HepG2 cells were seeded into 96-well plates at a density of 5 × 10^4^ cells/well and allowed to stabilize overnight. Then, the cells were incubated with increasing concentrations of jasmone (1–100 μg/mL) either alone or in combination with TCDD (13.5 nM) or FICZ (22.6 µM) for 24 h. After that, the medium was discarded, and the wells were washed once with sterile PBS and incubated with 50 µL of media with pro-luminescent substrate luciferin-6‘-chlorethylether (luciferin-CEE) in a final concentration of 100 µM at 37 °C for 3 h. Then, 25 µL of solution was transferred into a white 96-well plate, and 25 µL of detection reagent was added. The reaction mixture was incubated for 20 min in the dark. Luciferase activity was measured using a Tecan Infinite M200 plate luminometer (Schoeller Instruments, Praha-Kunratice, Czech Republic). Experiments were performed in at least three consecutive cell passages, and the treatments were conducted in triplicate (technical replicates).

### 4.12. Statistical Analyses

Data were analyzed using GraphPad Prism Version 6 (GraphPad Software, San Diego, CA, USA). The differences between the tested samples and either the negative or positive controls were tested using one-way ANOVA with Dunnett’s post hoc test. Significantly different values were marked with an asterisk.

## Figures and Tables

**Figure 1 ijms-24-15655-f001:**
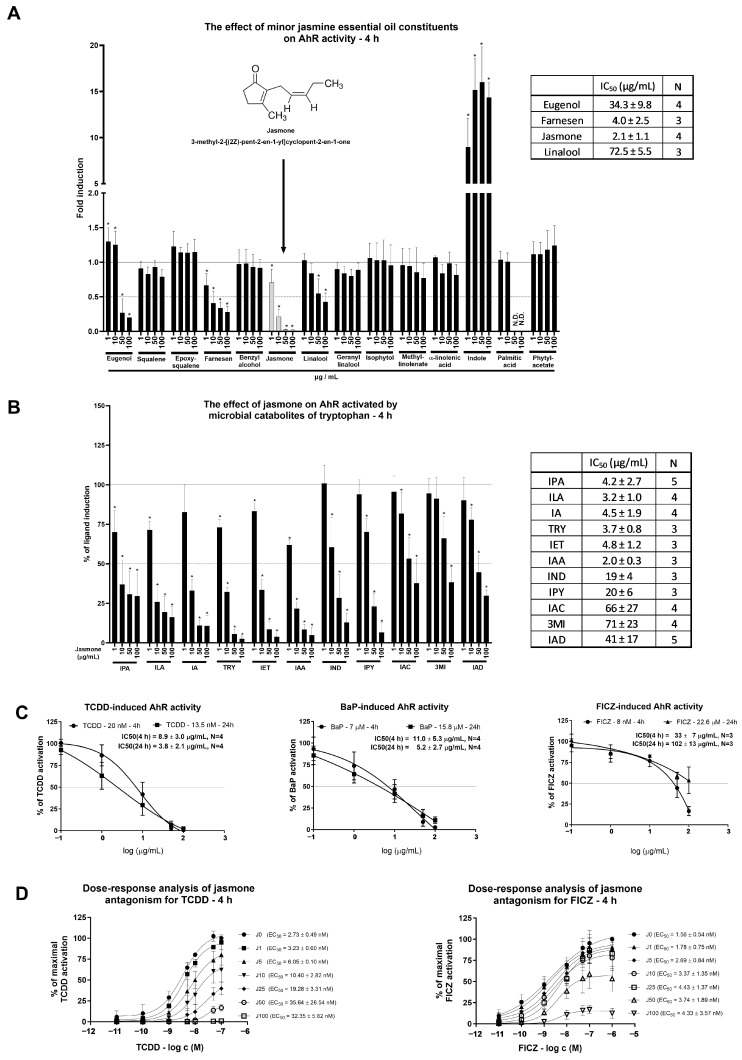
AZ-AHR cells were incubated either alone (negative control, set to 1, absent in the figure) or with selected minor constituents (1–10%) of jasmine essential oil in concentrations ranging from 1 to 100 µg/mL for 4 h (**A**). Microbial catabolites of tryptophan (MICTs), namely indole-3-propionic acid (IPA), indole-3-lactic acid (ILA), indole-3-aldehyde (IA), tryptamine (TRY), indole-3-ethanol (IET), indole-3-acetic acid (IAA), indole (IND), indole-3-pyruvate (IPY), indole-3-acrylate (IAC), 3-methylindole (3MI), and indole-3-acetamide (IAD), were incubated at a concentration of 10 µM with jasmone (1–100 µg/mL) in DMEM-depleted medium for 4 h in the AZ-AHR cell line (**B**). Reporter cell line AZ-AHR was exposed to jasmone in the presence of model AhR agonists comprising TCDD (2,3,7,8-tetrachlorodibenzo-p-dioxin; 20 nM for 4 h; 13.5 nM for 24 h), BaP (Benzo[a]pyrene; 7 µM for 4 h; 15.8 µM for 24 h), and FICZ (6-Formylindolo [3,2-b]carbazole; 8 nM for 4 h; 22.6 µM for 24 h) (**C**). Dose-response analyses of TCDD (0.01–100 nM) and FICZ (0.01–1000 nM) with increasing jasmone concentrations (1–100 µg/mL) after 4 h in DMEM-depleted medium (**D**). Following the treatments, cells were lysed, and luciferase activity was measured as described in Section 4. The calculation of IC50 and EC50 was performed in the GraphPad Prism software (version 10.0.3). Data are expressed as % of induction by the highest concentration of the appropriate ligand. Data are averages of 3 or more independent cell passages. * represents a significantly different value (*p* < 0.05) compared to the negative control (**A**) or ligand alone (**B**).

**Figure 2 ijms-24-15655-f002:**
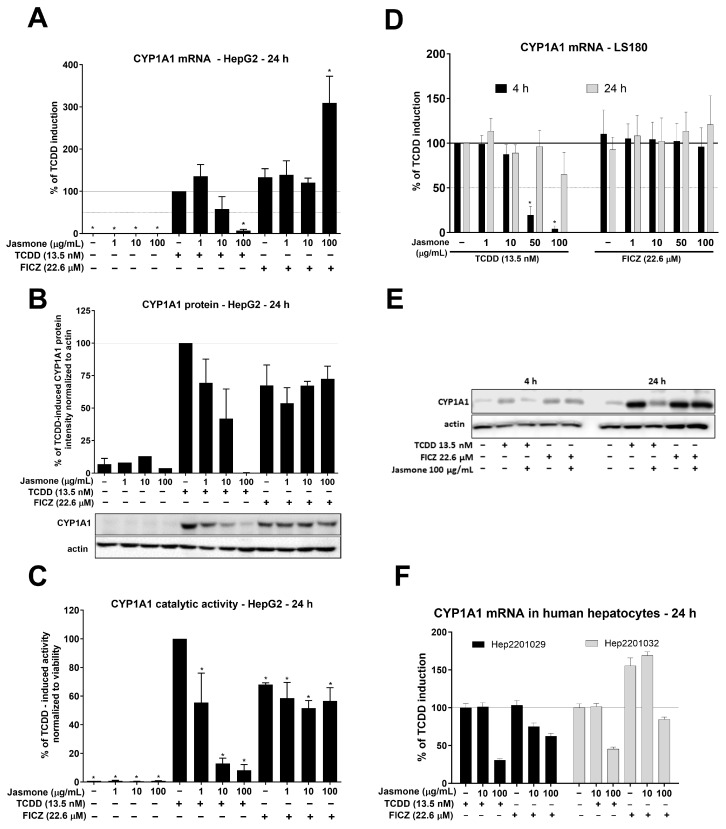
HepG2 cells were treated for 24 h with increasing concentrations of jasmone in the absence or presence of TCDD or FICZ, and CYP1A1 mRNA (**A**), protein (**B**), and catalytic activity (**C**) were determined as described in Section 4. LS180 cells were treated for 4 or 24 h with jasmone in the presence of TCDD or FICZ, and CYP1A1 mRNA (**D**) and protein (**E**) were determined. (**F**) Primary cultures of human hepatocytes obtained from two different liver tissue donors were treated with jasmone (10, 100 µg/mL) and TCDD (10 nM) or FICZ (22.6 µM) for 24 h. The bar graph shows a fold induction of CYP1A1 mRNA over the control cells. Data are expressed as % of TCDD induction. * represents a significantly different value (*p* < 0.05) compared to the pure lysate (**A**) or negative control (**B**,**C**).

**Figure 3 ijms-24-15655-f003:**
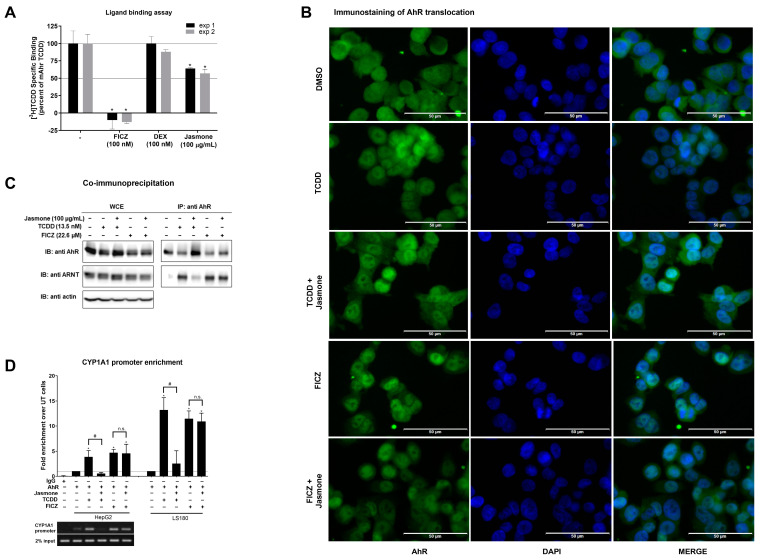
(**A**) Cytosolic extract from Hepa1c1c7 cells was incubated with vehicle (negative control; 0.1% *v/v*), FICZ (positive control; 100 nM), DEX (dexamethasone; non-specific binding; 100 nM), and jasmone (100 µg/mL) in the presence of 2 nM [^3^H]-TCDD. The specific binding of [^3^H]-TCDD was determined as a difference between total and non-specific (200 nM; TCDF) reactions (value for vehicle DMSO; 0.1% *v/v* = corresponds to the specific binding of [^3^H]-TCDD = 100%). Two independent experiments were performed, and the incubations were carried out in triplicate in each experiment (technical replicates). The error bars represent the mean ± SD. * represents a value significantly different from that of the vehicle (*p* < 0.05). (**B**) Sub-cellular localization of AhR in LS180 cells was mo-nitored as described in detail in Section 4.4. Representative micrographs are shown. Size bars inserted in individual pictures are equal to 50 µm. (**C**) Heterodimerization of AhR with ARNT. Protein immunoprecipitation—formation of AhR-ARNT heterodimer in LS180 cells incubated for 90 min with vehicle (DMSO; 0.1% *v*/*v*), TCDD (13.5 nM), and FICZ (22.6 µM) alone or with jasmone (100 µg/mL). Representative immunoblots (IBs) of immunoprecipitated protein (IP) eluates and whole-cell extract (WCE) are shown. Experiments were performed in three consecutive cell passages. (**D**) Chromatin immunoprecipitation (ChIP)—binding of AhR to CYP1A1 promoter in HepG2 or LS180 cells treated for 90 min with TCDD (13.5 nM) or FICZ (22.6 µM) alone, or with jasmone (100 µg/mL). Bar graphs show the enrichment of CYP1A1 promoter with AhR compared to vehicle-treated cells. *, #—represent significantly different result from negative control (un-treated sample) or positive control (TCDD), respectively. n.s.—represents non-significant result from positive control (FICZ). Representative DNA fragments from HepG2 cells amplified using PCR and analyzed on 2% agarose gel are shown. Experiments were performed in three consecutive passages.

**Table 1 ijms-24-15655-t001:** List of primers with corresponding UPL probes used for PCR.

Name of the Gene	Primer Sequences (F/R)	UPL Number
CYP1A1	CCAGGCTCCAAGAGTCCA GATCTTGGAGGTGGCTGCT	33
CYP1A2	ACAACCCTGCCAATCTCAAG GGGAACAGACTGGGACAATG	34
CYP1B1	ACGTACCGGCCACTATCACT CTCGAGTCTGCACATCAGGA	20
AhRR	CAAAACCCAGAGCAGACACC CAGTTCCGATTCGCACAGA	77
GAPDH	CTCTGCTCCTCCTGTTCGAC ACGACCAAATCCGTTGACTC	60

## Data Availability

The data presented in this study are available online in this article.

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
