# Peer review of "Jasmone Is a Ligand-Selective Allosteric Antagonist of Aryl Hydrocarbon Receptor (AhR)"

_ijms, 2023, doi:10.3390/ijms242115655_

Round 1
Reviewer 1 Report
Comments and Suggestions for Authors
COMMENTS TO THE AUTHORS (MANUSCRIPT ijms-2646880)
In this manuscript, Vrzal et al. report a study on the phytoderivative jasmone as non-competitive antagonist of aryl hydrocarbon receptor (AhR).
In general terms, although the rationale is acceptable and the central objective of the article is clear, the experimental conditions are very complex (cell lines, incubation times, concentrations, etc.), written in a very complex way, which results in a hard reading that misses the objective of the work. Then, there are issues which need to be addressed by the authors, rewriting the results section with reorganized figures, making them more simples.
In particular, receptor activation involves an ordered succession of steps (ligand binding, translocation to the nucleus, dimerization, DNA binding, etc.) that must be present when analyzing the results and in accordance with what has been described.
The use of MTT reduction as a cell viability assay. It is not correct the use of only one wavelength to measure MTT reduction. Experiments have to be performed using two wavelengths: one to measure blue coloration (formazan, depending of the solvent; if formazan is dissolved in DMSO the appropriate wavelength is approx. 500 nm; authors use 570 nm) and a second wavelength for background (usually at 650-670 nm). Figures have to plot the difference between both values. This analysis permits the accurate comparison between different samples.
Minor points:
Lines 140-142, the results described are interchanged with what is shown in the figure.
Lines 149-150, compare the results of figure 1D, but left panel x axis does not show values at concentration -6 (-loc C).
Line 228, “TCDF” ?
Author Response
COMMENTS TO THE AUTHORS (MANUSCRIPT ijms-2646880)
In this manuscript, Vrzal et al. report a study on the phytoderivative jasmone as non-competitive antagonist of aryl hydrocarbon receptor (AhR).
Comment:
In general terms, although the rationale is acceptable and the central objective of the article is clear, the experimental conditions are very complex (cell lines, incubation times, concentrations, etc.), written in a very complex way, which results in a hard reading that misses the objective of the work. Then, there are issues which need to be addressed by the authors, rewriting the results section with reorganized figures, making them more simples.
Reply to the comment:
We do appreciate the reviewer’s suggestions. However, it is difficult to make any reasonable changes without being more specific. The primary objective of the work was to characterize jasmone activity towards AhR.
Comment:
In particular, receptor activation involves an ordered succession of steps (ligand binding, translocation to the nucleus, dimerization, DNA binding, etc.) that must be present when analyzing the results and in accordance with what has been described.
Reply to the comment:
We agree with the reviewer; exactly this ordered succession of steps was described in the Introduction section. Moreover, each step was evaluated throughout the manuscript. The results section describes first the end-points (reporter gene assay, PCR) because these methods serve as a conversion funnel for any effect worth further investigation.
Comment:
The use of MTT reduction as a cell viability assay. It is not correct the use of only one wavelength to measure MTT reduction. Experiments have to be performed using two wavelengths: one to measure blue coloration (formazan, depending of the solvent; if formazan is dissolved in DMSO the appropriate wavelength is approx. 500 nm; authors use 570 nm) and a second wavelength for background (usually at 650-670 nm). Figures have to plot the difference between both values. This analysis permits the accurate comparison between different samples.
Reply to the comment:
We agree with the reviewer that two wavelengths may bring certain benefits under specific circumstances. However, according to our experience and general recommendations for measuring MTT assay, a maximum absorption wavelength is around 570nm, even in DMSO as solvent. Moreover, the absorbance of samples correlates with our visible observations, i.e. a decline in absorbance in a culture dish is reflected in the decline in the absorbance for a given well. Thus, we are not convinced that measuring the second wavelength would change the main output, i.e. antagonistic activity of jasmone for AhR, and weak inhibition in MTT assay.
Comment:
Minor points:
Lines 140-142, the results described are interchanged with what is shown in the figure.
Reply to the comment:
We are not sure, what the reviewer has on mind. Line 140 is the end of Figure legend and it states what was used for comparison to get significance.
Comment:
Lines 149-150, compare the results of figure 1D, but left panel x axis does not show values at concentration -6 (-loc C).
Reply to the comment:
Yes, that is true. As it is stated in the figure legend of Figure 1, subfigure D, TCDD was tested up to 100nM (i.e. 10-7 M = -7 when using the logarithm), while FICZ was tested up to 1000nM (i.e. 10-6 M = -6 when using logarithm). Therefore, no values exist at concentration -6 of the left Figure 1D.
Comment:
Line 228, “TCDF” ?
Reply to the comment:
TCDF = 2,3,7,8-tetrachlorodibenzofuran, a compound used for subtraction of non-specific binding for the purposes of ligand binding assay. It is mentioned in Materials and Mehods.
Reviewer 2 Report
Comments and Suggestions for Authors
The paper of Vrzal et al. is devoted to study of jasmone as non-competitive antagonist of aryl hydrocarbon receptor (AhR). The authors have carried out the screening of minor constituents of jasmine essential oil towards aryl hydrocar-bon receptor (AhR) activity and found that jasmone efficiently inhibited AhR activation. Further, jasmone-mediated inhibition of AhR was studied in details.
The study is well-designed and described. However, for my opinion, several point should be addressed.
In Materials and Methods section, the description of AZ-AHR cells is missing;
Figure 1. What is a control for these experiments? The authors show a “Fold induction”. Related what? What is taken as a unit?
The authors should provide explanation why they have been used LS180 cell line in experiments.
Figure 2F. How do authors explain jasmone-mediated inhibition of response to FICS in human hepatocytes but not in Hep2G and LS180? Is this Figure necessary?
Figure 3B. TCDD-induced cytoplasmic degradation of AhR is clearly seen from this Figure. This was abolished by jasmone which points to jasmone-mediated interference with AhR stability regulation.
What is about jasmone-mediated impact on AhR stability and translocation in HepG2 cells?
For my opinion, if the authors make a clear emphasis on the difference between AhR-mediated signaling events (it’s stability or/and translocation) in the case of TCDD- and TCDD/jasmone stimulation, this will prove the paper’s line.
The authors may use the experiment on the check of AhR stability with cycloheximide under TCDD and TCDD/jasmone stimulation.
The paper will benefit if the authors describe the signaling events under AhR activation and turnover as well as the regulation of AhR stability and signaling properties.
CHIP assay which was carried out by the authors significantly enhanced all given conclusions.
Comments on the Quality of English LanguageEnglish language is fine or minor editing is required
Author Response
The paper of Vrzal et al. is devoted to study of jasmone as non-competitive antagonist of aryl hydrocarbon receptor (AhR). The authors have carried out the screening of minor constituents of jasmine essential oil towards aryl hydrocar-bon receptor (AhR) activity and found that jasmone efficiently inhibited AhR activation. Further, jasmone-mediated inhibition of AhR was studied in details.
The study is well-designed and described. However, for my opinion, several point should be addressed.
Comment:
In Materials and Methods section, the description of AZ-AHR cells is missing;
Reply to the comment:
We do apologize, a citation, where characterization of this cell line was performed, was added to section 4.7.
Comment:
Figure 1. What is a control for these experiments? The authors show a “Fold induction”. Related what? What is taken as a unit?
Reply to the comment:
We do apologize. The control for experiments in Figure 1A was negative control, untreated cells, arbitrarily set to 1. We added this information to the legend of Figure 1.
Comment:
The authors should provide explanation why they have been used LS180 cell line in experiments.
Reply to the comment:
The explanation was added to the results section, line 161.
Comment:
Figure 2F. How do authors explain jasmone-mediated inhibition of response to FICS in human hepatocytes but not in Hep2G and LS180? Is this Figure necessary?
Reply to the comment:
Yes, we believe this figure is necessary since it demonstrates jasmone action in the only physiologic in vitro model used within this study, primary culture of human hepatocytes (PCHH). The rest of the study used easy-to-cultivate cancerous cells, which usually lack a biotransformation apparatus, which is inducible.
The reason why jasmone antagonized FICZ much more effectively in PCHH may (among other reasons) reflect metabolism. In contrast to hepatocarcinoma cells, HepG2 or adenocarcinoma cells, LS180, which do not constitutively express CYP1A2, this is true for PCHH. Since CYP1A1 was demonstrated to be inhibited by nanomolar concentrations of FICZ (Dong et al., 2023, 10.1177/11786469231182508), the reasonable explanation lies in the potent inhibition of low or even induced level of CYP1A1 in cancer cell lines. This does not apply for PCHH, where constitutively and abundantly expressed CYP1A2 may contribute to FICZ metabolism since the very beginning of the treatment. This explanation was added to the manuscript to the Discussion section.
However, other mechanisms may lie beyond our observation as well.
Comment:
Figure 3B. TCDD-induced cytoplasmic degradation of AhR is clearly seen from this Figure. This was abolished by jasmone which points to jasmone-mediated interference with AhR stability regulation.
Reply to the comment:
We do not fully understand this comment. In Figure 3B, the loss of AhR from cytoplasmic compartment as well as increase in nuclear compartment is observed, as classical signaling AhR defines. The message of this figure is, that jasmone has no impact on ligand-induced translocation.
Comment:
What is about jasmone-mediated impact on AhR stability and translocation in HepG2 cells?
For my opinion, if the authors make a clear emphasis on the difference between AhR-mediated signaling events (it’s stability or/and translocation) in the case of TCDD- and TCDD/jasmone stimulation, this will prove the paper’s line.
Reply to the comment:
This is very interesting issue. However, by performing western blot analysis, we personally observed that jasmone alone did not affect AhR protein level, i.e. stability in LS180 cells. The impact of jasmone on ligand-triggered degradation of AhR (as observed particularly in Figure S5) is apparently dependent on the nature of ligand. Moreover, it can be speculated that jasmone affects other molecular targets implicated in AhR signaling.
Comment:
The authors may use the experiment on the check of AhR stability with cycloheximide under TCDD and TCDD/jasmone stimulation.
Reply to the comment:
We thank the reviewer for this idea. For better understanding what possible other molecular targets jasmone may have, cycloheximide chase assay would be suitable. However, for the purpose of this study to demonstrate the impact of jasmone on AhR signaling, we believe our data are sufficient.
Comment:
The paper will benefit if the authors describe the signaling events under AhR activation and turnover as well as the regulation of AhR stability and signaling properties.
Reply to the comment:
We thank the reviewer for interesting comment. The basic concept of AhR signaling was described in the 2nd paragraph of the Introduction section.
Comment:
CHIP assay which was carried out by the authors significantly enhanced all given conclusions.
Reply to the comment:
We thank the reviewer for this comment.
Comment:
English language is fine or minor editing is required
Reply to the comment:
Based on reviewer's recommendation, we used the software “Grammarly” to check the English language in the manuscript.
Round 2
Reviewer 1 Report
Comments and Suggestions for Authors
COMMENTS TO THE AUTHORS (MANUSCRIPT ijms-2646880) V2
In this manuscript, Vrzal et al. report a study on the phytoderivative jasmone as non-competitive antagonist of aryl hydrocarbon receptor (AhR).
Comment:
In general terms, although the rationale is acceptable and the central objective of the article is clear, the experimental conditions are very complex (cell lines, incubation times, concentrations, etc.), written in a very complex way, which results in a hard reading that misses the objective of the work. Then, there are issues which need to be addressed by the authors, rewriting the results section with reorganized figures, making them more simples.
Reply to the comment:
We do appreciate the reviewer’s suggestions. However, it is difficult to make any reasonable changes without being more specific. The primary objective of the work was to characterize jasmone activity towards AhR.
New comment:
I really think it can be done an effort to reconsider the writing of results section.
Comment:
In particular, receptor activation involves an ordered succession of steps (ligand binding, translocation to the nucleus, dimerization, DNA binding, etc.) that must be present when analyzing the results and in accordance with what has been described.
Reply to the comment:
We agree with the reviewer; exactly this ordered succession of steps was described in the Introduction section. Moreover, each step was evaluated throughout the manuscript. The results section describes first the end-points (reporter gene assay, PCR) because these methods serve as a conversion funnel for any effect worth further investigation.
New comment:
I disagree with authors that “each step was evaluated throughout the manuscript”.
Comment:
The use of MTT reduction as a cell viability assay. It is not correct the use of only one wavelength to measure MTT reduction. Experiments have to be performed using two wavelengths: one to measure blue coloration (formazan, depending of the solvent; if formazan is dissolved in DMSO the appropriate wavelength is approx. 500 nm; authors use 570 nm) and a second wavelength for background (usually at 650-670 nm). Figures have to plot the difference between both values. This analysis permits the accurate comparison between different samples.
Reply to the comment:
We agree with the reviewer that two wavelengths may bring certain benefits under specific circumstances. However, according to our experience and general recommendations for measuring MTT assay, a maximum absorption wavelength is around 570nm, even in DMSO as solvent. Moreover, the absorbance of samples correlates with our visible observations, i.e. a decline in absorbance in a culture dish is reflected in the decline in the absorbance for a given well. Thus, we are not convinced that measuring the second wavelength would change the main output, i.e. antagonistic activity of jasmone for AhR, and weak inhibition in MTT assay.
New comment:
I would like to see an spectra of MTT solution after addition to the cultures to demonstrate that “a maximum absorption wavelength is around 570nm, even in DMSO as solvent.” Simply, I have never seen that maximum.
Comment:
Minor points:
Lines 140-142, the results described are interchanged with what is shown in the figure.
Reply to the comment:
We are not sure, what the reviewer has on mind. Line 140 is the end of Figure legend and it states what was used for comparison to get significance.
New comment:
According to the file I received, lines 140-142 state: “FICZ-induced luciferase activity was affected less intensively after 4 hrs (IC50 = 33 ± 7 μg/mL) and almost disappeared after 24 hrs (IC50 = 102 ± 13 μg/mL).”, not the end of figure legend as indicated by the authors. And I still consider that the results described are interchanged with what is shown in the figure 1C. I do not rule out being wrong, but data is confusing.
Comment:
Lines 149-150, compare the results of figure 1D, but left panel x axis does not show values at concentration -6 (-loc C).
Reply to the comment:
Yes, that is true. As it is stated in the figure legend of Figure 1, subfigure D, TCDD was tested up to 100nM (i.e. 10-7 M = -7 when using the logarithm), while FICZ was tested up to 1000nM (i.e. 10-6 M = -6 when using logarithm). Therefore, no values exist at concentration -6 of the left Figure 1D.
New comment:
Well, I consider the data at concentration -6 important. The curves have not reached their maximum.
Comment:
Line 228, “TCDF” ?
Reply to the comment:
TCDF = 2,3,7,8-tetrachlorodibenzofuran, a compound used for subtraction of non-specific binding for the purposes of ligand binding assay. It is mentioned in Materials and Mehods.
New comment:
Okey
Author Response
COMMENTS TO THE AUTHORS (MANUSCRIPT ijms-2646880) V2
In this manuscript, Vrzal et al. report a study on the phytoderivative jasmone as non-competitive antagonist of aryl hydrocarbon receptor (AhR).
Comment:
In general terms, although the rationale is acceptable and the central objective of the article is clear, the experimental conditions are very complex (cell lines, incubation times, concentrations, etc.), written in a very complex way, which results in a hard reading that misses the objective of the work. Then, there are issues which need to be addressed by the authors, rewriting the results section with reorganized figures, making them more simples.
Reply to the comment:
We do appreciate the reviewer’s suggestions. However, it is difficult to make any reasonable changes without being more specific. The primary objective of the work was to characterize jasmone activity towards AhR.
New comment:
I really think it can be done an effort to reconsider the writing of results section.
Reply to the new comment:
The results section was rewritten as requested.
Comment:
In particular, receptor activation involves an ordered succession of steps (ligand binding, translocation to the nucleus, dimerization, DNA binding, etc.) that must be present when analyzing the results and in accordance with what has been described.
Reply to the comment:
We agree with the reviewer; exactly this ordered succession of steps was described in the Introduction section. Moreover, each step was evaluated throughout the manuscript. The results section describes first the end-points (reporter gene assay, PCR) because these methods serve as a conversion funnel for any effect worth further investigation.
New comment:
I disagree with authors that “each step was evaluated throughout the manuscript”.
Reply to the new comment:
At this point, we have to disagree with the opponent. The complete signaling cascade of AhR was evaluated, including ligand binding (Fig 3A), translocation (Fig 3B), dimerization (Fig 3C), DNA binding (Fig 3D), and target genes expression (Fig 2).
If any relevant step in AhR signaling is missing, we kindly ask the reviewer what step it is to meet the reviewer’s requirements.
Comment
The use of MTT reduction as a cell viability assay. It is not correct the use of only one wavelength to measure MTT reduction. Experiments have to be performed using two wavelengths: one to measure blue coloration (formazan, depending of the solvent; if formazan is dissolved in DMSO the appropriate wavelength is approx. 500 nm; authors use 570 nm) and a second wavelength for background (usually at 650-670 nm). Figures have to plot the difference between both values. This analysis permits the accurate comparison between different samples.
Reply to the comment:
We agree with the reviewer that two wavelengths may bring certain benefits under specific circumstances. However, according to our experience and general recommendations for measuring MTT assay, a maximum absorption wavelength is around 570nm, even in DMSO as solvent. Moreover, the absorbance of samples correlates with our visible observations, i.e. a decline in absorbance in a culture dish is reflected in the decline in the absorbance for a given well. Thus, we are not convinced that measuring the second wavelength would change the main output, i.e. antagonistic activity of jasmone for AhR, and weak inhibition in MTT assay.
New comment:
I would like to see an spectra of MTT solution after addition to the cultures to demonstrate that “a maximum absorption wavelength is around 570nm, even in DMSO as solvent.” Simply, I have never seen that maximum.
Reply to the new comment:
We followed standard MTT viability protocol provided by multiple manufacturers, e.g. https://www.thermofisher.com/order/catalog/product/V13154.
Moreover, we measured spectra of purple-colored formazan (as a product of MTT) in DMSO with spectrophotometer TECAN, M200Pro (see figure below). It can be seen that absorption maximum is around 570nm.
Figure legend: LS180 cells were treated with MTT at 1 mg/mL in DMEM medium for 1 h. Then, the medium was discarded, and insoluble purple formazan was dissolved in DMSO. The absorption spectrum was measured from 380-780nm with a step of 2 nm. The results are an average of two wells.
Comment:
Minor points:
Lines 140-142, the results described are interchanged with what is shown in the figure.
Reply to the comment:
We are not sure, what the reviewer has on mind. Line 140 is the end of Figure legend and it states what was used for comparison to get significance.
New comment:
According to the file I received, lines 140-142 state: “FICZ-induced luciferase activity was affected less intensively after 4 hrs (IC50 = 33 ± 7 μg/mL) and almost disappeared after 24 hrs (IC50 = 102 ± 13 μg/mL).”, not the end of figure legend as indicated by the authors. And I still consider that the results described are interchanged with what is shown in the figure 1C. I do not rule out being wrong, but data is confusing.
Reply to the new comment:
We do apologize, the description stated in those lines refers to TCDD and BaP from previous sentence. We do agree that it is confusing when focused solely on FICZ after 4 and 24 hours. The sentence was modified.
Comment:
Lines 149-150, compare the results of figure 1D, but left panel x axis does not show values at concentration -6 (-loc C).
Reply to the comment:
Yes, that is true. As it is stated in the figure legend of Figure 1, subfigure D, TCDD was tested up to 100nM (i.e. 10-7 M = -7 when using the logarithm), while FICZ was tested up to 1000nM (i.e. 10-6 M = -6 when using logarithm). Therefore, no values exist at concentration -6 of the left Figure 1D.
New comment:
Well, I consider the data at concentration -6 important. The curves have not reached their maximum.
Reply to the new comment:
Unfortunately, we do not have this data as the concentration range for TCDD ended at 100 nM (10-7 M). Moreover, we repeatedly observed (our recent paper: https://doi.org/10.1038/s41467-023-38478-6), that we reached the plateau at 4hrs for 100nM TCDD as it can be observed for dark-filled circles representing TCDD without jasmone.
Since we repeatedly observed very similar GraphPad software-calculated inter-experimental EC50s, we had no reason to add a higher concentration.
Comment:
Line 228, “TCDF” ?
Reply to the comment:
TCDF = 2,3,7,8-tetrachlorodibenzofuran, a compound used for subtraction of non-specific binding for the purposes of ligand binding assay. It is mentioned in Materials and Mehods.
New comment:
Okey
Reply to the new comment: Thank you.

Reviewer 2 Report
Comments and Suggestions for Authors
The authors have addrssed the majority of issues required
Author Response
Comments and Suggestions for Authors
The authors have addrssed the majority of issues required
Reply:
We thank the reviewer for suggestions and it was a pleasure to fill the recommendations.
Round 3
Reviewer 1 Report
Comments and Suggestions for Authors
None